# MULTI-LABEL KNOWLEDGE DISTILLATION

## ABSTRACT

Existing knowledge distillation methods typically work by enforcing the consistency of output logits or intermediate feature maps between the teacher network and student network. Unfortunately, these methods can hardly be extended to the multi-label learning scenario. Because each instance is associated with multiple semantic labels, neither the prediction logits nor the feature maps obtained from the whole example can accurately transfer knowledge for each label. In this paper, we propose a novel multi-label knowledge distillation method. On one hand, it exploits the informative semantic knowledge from the logits by label decoupling with the one-versus-all reduction strategy; on the other hand, it enhances the distinctiveness of the learned feature representations by leveraging the structural information of label-wise embeddings. Experimental results on multiple benchmark datasets validate that the proposed method can avoid knowledge counteraction among labels, and achieve superior performance against diverse comparing methods.

## 1 INTRODUCTION

Despite the remarkable success in training deep neural networks (DNNs) (Krizhevsky et al., 2012), it is hard to deploy these large neural networks on lightweight terminals, *e.g.*, mobile phones, under the constraint of computational resource or requirement of short inference time. To mitigate this issue, knowledge distillation (Hinton et al., 2015) aims to improve the performance of a small network (also known as the student) by requiring the knowledge from a large network (also known as the teacher) to guide the training of the student network. Existing knowledge distillation methods can be roughly divided into two categories: logits-based methods and feature-based methods. The former minimizes the difference between logits of teacher model and student model (Hinton et al., 2015; Zhao et al., 2022), while the latter distills knowledge from feature maps of intermediate layers (Park et al., 2019; Tian et al., 2019; Chen et al., 2021).

Typical knowledge distillation methods focus on the multi-class classification task, where each instance is associated with only one class label. However, in many real-world scenarios, an instance inherently contains complex semantics and can be simultaneously assigned with multiple class labels. For example, an image of street scene may be annotated with labels *building*, *car* and *person*. To learn the complex object-label mapping, there is always necessity of training large models to obtain desirable performance in multi-label classification. Unfortunately, due to computational resource constraints, it cannot be allowed to adopt large neural networks in many practical applications, leading to noticeable decrease in model performance (Gou et al., 2021). To alleviate the performance degradation, it is necessary to design specific knowledge distillation methods for multi-label learning. We formalize such a problem as a new learning framework called Multi-Label Knowledge Distillation (MLKD).

Although knowledge distillation has been proven to be effective for improving the performance of the student network in single-label classification, it is still a challenging problem to directly extend existing KD methods to solve MLKD problems. Specifically, logits-based methods often obtain the predicted probabilities based on the softmax function, which is unavailable for MLKD, since the sum of predicted probabilities may not equal to one in Multi-Label Learning (MLL). Feature-based methods often perform knowledge distillation based on feature maps of the whole image with multiple semantics, which makes the model focus on the major objects while neglect the minor objects. This would lead the model to obtain sub-optimal even undesirable distillation performance. Figure 1 provides empirical validations for these observations.

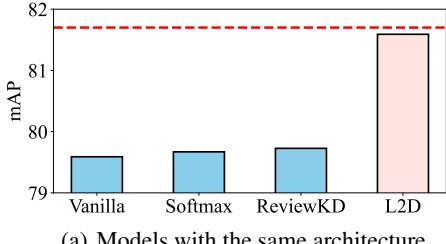 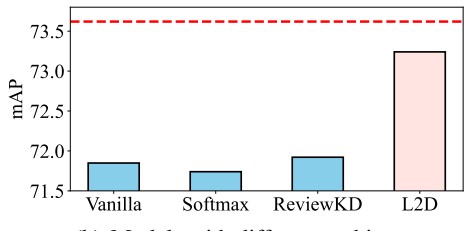

(a) Models with the same architecture    (b) Models with different architectures

Figure 1: Comparison results between our proposed L2D method and conventional KD methods on MS-COCO. We compare our method with the following three baselines: 1) Vanilla: student trained without distillation; 2) Softmax (Hinton et al., 2015): a representative logits-based method by measuring KL divergence on softmax score after logits; 3) ReviewKD (Chen et al., 2021): a feature-based method that achieves sota performance. The red dashed lines mark the performance of teachers. Conventional KD methods achieve unfavorable performance similar to the student, while our method significantly outperforms these methods and achieves comparable performance to the teacher. Details about the implementation of this experiment can be found in Appendix A.

In this paper, to perform multi-label knowledge distillation, we propose a new method consisting of multi-label logits distillation and label-wise embedding distillation (L2D for short). Specifically, to exploit informative semantic knowledge compressed in the logits, L2D employs the one-versus-all reduction strategy to obtain a series of binary classification problems and perform logits distillation for each one. To enhance the distinctiveness of learned feature representations, L2D encourages the student model to maintain a consistent structure of intra-class and intra-instance (inter-class) label-wise embeddings with the teacher model. By leveraging the structural information of the teacher model, these two structural consistencies respectively enhance the compactness of intra-class embeddings and dispersion of inter-class embeddings for the student model.

Our main contributions can be summarized as follows:

- A general learning framework called MLKD is proposed. To our best knowledge, the framework is the first study specially designed for knowledge distillation in the multi-label learning scenario.
- A new approach called L2D is proposed. It performs multi-label logits distillation and label-wise embedding distillation simultaneously. The former provides informative semantic knowledge while the latter encourages the student model to learn more distinctive feature representations.
- Extensive experimental results on benchmark datasets demonstrate the effectiveness of our proposed method.

## 2   RELATED WORK

The concept of knowledge distillation (KD) proposed by Hinton et al. (2015) defines a learning framework that transfers knowledge from a large teacher network to a small student network. Existing works can be roughly divided into two groups, *i.e.*, logits-based methods and feature-based methods. Logits-based methods mainly focus on designing effective distillation losses to distill knowledge from logits and softmax scores after logits. DML (Zhang et al., 2018) introduces a mutual learning method to train both teachers and students simultaneously. TAKD (Mirzadeh et al., 2020) proposes a new architecture called "teacher assistant", which is an intermediate-sized network bridging the gap between teachers and students. Besides, a recent study (Zhao et al., 2022) proposes a novel logits-based method to reformulate the classical KD loss into two parts and achieves the state-of-the-art performance by adjusting weights for these two parts. Some other methods focus on distilling knowledge from intermediate feature layers. FitNet (Romero et al., 2014) is the first approach to distill knowledge from intermediate features by measuring the distance between feature maps. Attention transfer (Zagoruyko & Komodakis, 2016a) achieves better performance than FitNet

by distilling knowledge from the attention maps. PKT (Passalis & Tefas, 2018) measures the KL divergence between features by treating them as probability distributions. RKD (Park et al., 2019) utilizes the relations among instances to guide the training process of the student model. CRD (Tian et al., 2019) incorporates contrastive learning into knowledge distillation. ReviewKD (Chen et al., 2021) proposes a review mechanism which uses multiple layers in the teacher to supervise one layer in the student. ITRD (Miles et al., 2021) aims to maximize the correlation and mutual information between the students' and teachers' representations.

Multi-label learning has increasingly attracted a lot of interest recently. Existing solutions for solving MLL problems can be categorized into three directions. The first type attempts to design novel loss functions for tackling the intrinsic positive-negative imbalance issue in multi-label classification tasks. For example, ASL (Ridnik et al., 2021a) uses different weights to re-weight positive and negative examples for the balanced training. The second type focuses on modeling the label correlations, which provides prior knowledge for multi-label classification. Among them, MLGCN (Chen et al., 2019b) is a representative method that employs a graph convolutional network to model correlation matrix. CADM (Chen et al., 2019a) constructs a similar graph based on class-aware maps. To handle the multiple semantic objects contained in an image, the last type of methods aims to locate areas of interest related to semantic labels by using attention techniques. Among them, C-Tran (Lanchantin et al., 2021) first utilizes the transformer to retrieve embeddings from visual features for each label. Query2Label (Liu et al., 2021a) uses several stacked transformer encoders to identify interesting areas. ML-Decoder (Ridnik et al., 2021b) simplifies transformer encoders in Query2Label. ADDS (Xu et al., 2022b) introduces encoders from CLIP (Radford et al., 2021) in order to get better textual and visual embedding inputs to the classification head. In addition, ADDS adds a multi-head cross-attention layer and a skipping connection from the query input to the query output based on ML-Decoder.

Several previous studies have applied KD techniques to improve the performance of MLL. For example, Liu et al. (2018) and Xu et al. (2022a) simply minimized mean squared error (MSE) loss between teacher logits and student logits. Song et al. (2021) designed a partial softmax function by combining a positive label with all other negative labels. Then, the conventional KD loss can be computed for each positive label. The main difference between our work and previous works is that we aim to improve the distillation performance instead of the performance of MLL. This can be reflected in experiments in Section 4 and Appendix B, where the proposed method mainly compares with KD methods while the previous method mainly compares with MLL methods.

## 3 THE PROPOSED APPROACH

Let $\boldsymbol{x} \in \mathcal{X}$ be an instance and $\boldsymbol{y} \in \mathcal{Y}$ be its corresponding label vector, where $\mathcal{X} \subset \mathbb{R}^d$ is the input space with $d$ dimensions and $\mathcal{Y} \subset \{0, 1\}^q$ is the target space with $q$ class labels. We further use $y_j$ to denote the $j$-th component of $\boldsymbol{y}$. For a given instance $\boldsymbol{x}$, $y_j = 1$ indicates the $j$-th label is relevant to the instance; $y_j = 0$, otherwise. In multi-label learning, each instance may be assigned with more than one label, which means $\sum_{j=1}^{q} \mathbb{I}(y_j = 1) \geq 1$, where $\mathbb{I}(\cdot)$ is the indicator function. We also denote by $[q]$ the integer set $\{1, 2, \cdots, q\}$.

In this paper, we use a classification model consisting of three components, *i.e.*, a visual backbone $f$, which extracts a feature map $f(\boldsymbol{x})$ for the input $\boldsymbol{x}$, a label-wise embedding encoder $\boldsymbol{g}$ (Lanchantin et al., 2021; Liu et al., 2021a), which produces a label-wise embedding $\boldsymbol{e}_k = g_k(f(\boldsymbol{x}))$ with respect to the $k$-th class based on the feature map $f(\boldsymbol{x})$, and a multi-label classifier $\boldsymbol{h}$, which predicts multi-label probabilities $\hat{\boldsymbol{y}} = [\sigma(h_1(\boldsymbol{e}_1)), \sigma(h_2(\boldsymbol{e}_2)), ..., \sigma(h_q(\boldsymbol{e}_q))]$, where $\sigma(\cdot)$ denotes the sigmoid function. It is noteworthy that the used model is very general, which can be built by equipping commonly used backbones, *e.g.*, ResNet (He et al., 2016), with a label-wise embedding encoder $\boldsymbol{g}$. For the notations mentioned above, we use the superscripts $\mathcal{T}$ (or $\mathcal{S}$) to denote the teacher (or student) model. For example, we use $\boldsymbol{e}_k^{\mathcal{T}}$ and $\boldsymbol{e}_k^{\mathcal{S}}$ to denote the label-wise embeddings for teacher and student models.

In multi-label learning, a popular method is to employ the one-versus-all reduction strategy to transform the original task into multiple binary problems. Among the various loss functions, the most commonly used one is the binary cross entropy (BCE) loss. Specifically, given a batch of examples

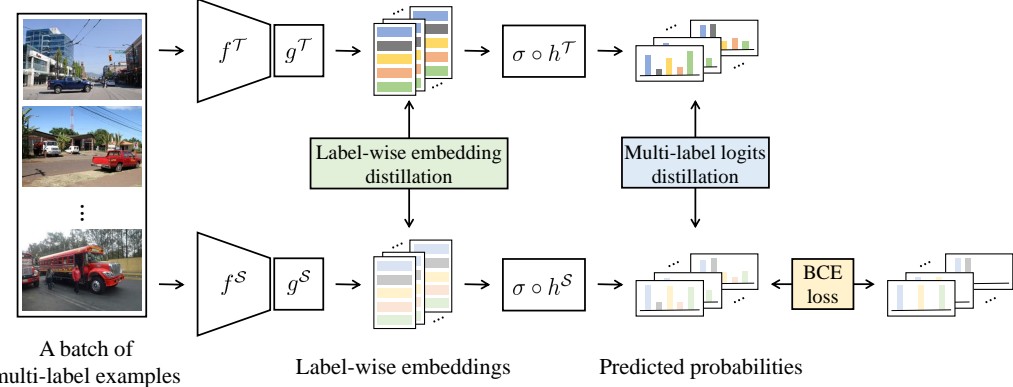

Figure 2: An illustration of the L2D framework. The framework simultaneously performs multi-label logits distillation and label-wise embedding distillation to improve the performance of the student model.

$\{(\boldsymbol{x}_i, \boldsymbol{y}_i)\}_{i=1}^{b}$ and the predicted probabilities $\hat{\boldsymbol{y}}$, the BCE loss can be defined as follows:

$$\mathcal{L}_{\text{BCE}} = -\frac{1}{b} \sum_{i=1}^{b} \sum_{k=1}^{q} y_{ik} \log(\hat{y}_{ik}) + (1 - y_{ik}) \log(1 - \hat{y}_{ik}). \tag{1}$$

Figure 2 illustrates the distillation process of the proposed L2D framework. For a batch of training examples, we feed them into the teacher/student model to obtain the label-wise embeddings and predicted probabilities. In order to train the student model, besides the BCE loss, we design the following two distillation losses: 1) multi-label logits distillation loss $\mathcal{L}_{\text{MLD}}$ to exploit informative semantic knowledge compressed in the logits, 2) label-wise embedding distillation loss $\mathcal{L}_{\text{LED}}$ to leverage structural information for enhancing the distinctiveness of learned feature representations. The overall objective function can be presented as

$$\mathcal{L}_{\text{L2D}} = \mathcal{L}_{\text{BCE}} + \lambda_{\text{MLD}} \mathcal{L}_{\text{MLD}} + \lambda_{\text{LED}} \mathcal{L}_{\text{LED}}, \tag{2}$$

where $\lambda_{\text{MLD}}$ and $\lambda_{\text{LED}}$ are two balancing parameters.

### 3.1 MULTI-LABEL LOGITS DISTILLATION

Traditional logits-based distillation normally minimizes the Kullback-Leibler (KL) divergence between the predicted probabilities, *i.e.*, the logits after the softmax function, of teacher and student model. However, the method cannot be directly applied to the MLL scenario, since it depends on a basic assumption that the predicted probabilities of all classes should sum to one, which hardly holds for MLL examples.

To mitigate this issue, inspired by the idea of one-versus-all reduction, we propose a multi-label logits distillation (MLD) loss, which decomposes the original multi-label task into multiple binary classification problems and minimizes the divergence between the binary predicted probabilities of two models. Formally, the MLD loss can be formulated as follows:

$$\mathcal{L}_{\text{MLD}} = \frac{1}{b} \sum_{i=1}^{b} \sum_{k=1}^{q} \mathcal{D}\left([\hat{y}_{ik}^{\mathcal{T}}, 1 - \hat{y}_{ik}^{\mathcal{T}}] \| [\hat{y}_{ik}^{\mathcal{S}}, 1 - \hat{y}_{ik}^{\mathcal{S}}]\right), \tag{3}$$

where $[\cdot, \cdot]$ is an operator used to concatenate two scalars into a vector, and $\mathcal{D}$ is a divergence function. The most common choice is the KL divergence $\mathcal{D}_{\text{KL}}(P\|Q) = \sum_{x \in \mathcal{X}} P(\boldsymbol{x}) \log\left(\frac{P(\boldsymbol{x})}{Q(\boldsymbol{x})}\right)$, where $P$ and $Q$ are two different probability distributions. The MLD loss aims to improve the performance of student model by sufficiently exploiting informative knowledge from logits.

### 3.2 LABEL-WISE EMBEDDING DISTILLATION

The MLD loss performs distillation on the predicted probabilities that can be regarded as a high-level representation, i.e., the final outputs of model. The knowledge distilled from the teacher model

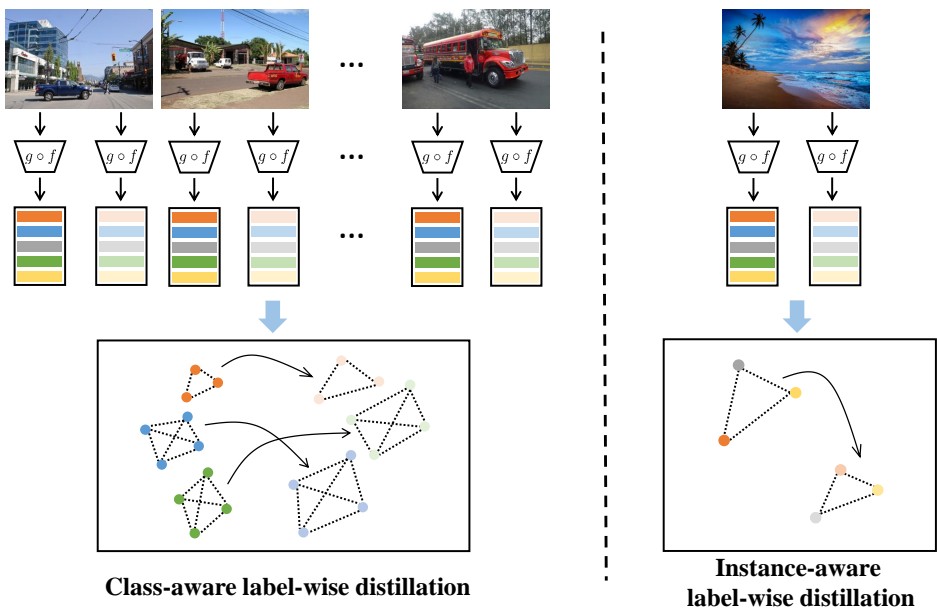

Figure 3: An illustration of class/instance-aware label-wise embedding distillation. Class-aware label-wise embedding distillation (CD) captures structural relations among intra-class label-wise embeddings from different examples, while instance-aware label-wise embedding distillation (ID) explores structural relations among intra-instance (inter-class) label-wise embeddings.

by only using the MLD loss would be insufficient to train a student model with desirable performance due to the limited information carried by the logits. To further strengthen the effectiveness of distillation, we design the label-wise embedding distillation (LED) loss, which aims to explore the structural knowledge from label-wise embeddings. The main idea is to capture two types of structural relations among label-wise embeddings: 1) class-aware label-wise embedding distillation (CD) loss $\mathcal{L}_{\text{LED-CD}}$, which captures the structural relation between any two intra-class label-wise embeddings from different examples; 2) instance-aware label-wise embedding distillation (ID) loss $\mathcal{L}_{\text{LED-ID}}$, which models the structural relation between any two inter-class label-wise embeddings from the same example. In the following content, we introduce these two distillation losses in detail.

### 3.2.1 CLASS-AWARE LABEL-WISE EMBEDDING DISTILLATION

Class-aware label-wise embedding distillation aims to improve the distillation performance by exploiting the structural relations among intra-class label-wise embeddings. Generally, the same semantic objects from two different images often differ from each other by their individual characteristics, such as two cars with different colors and diverse styles (see the left side in Figure 3). Since our goal is to distinguish between *car* and other semantic classes instead of identifying different cars, these distinctiveness would be confusing information for the corresponding classification task. Due to the powerful learning capacity, the large model is able to capture the highly abstract semantic representations for each class label by neglecting the useless individual information. From the perspective of learned feature representations, as shown in the left side of Figure 3, the teacher model tends to obtain a more compact structure of intra-class label-wise embedding, which often leads to better classification performance. By transferring the structural knowledge from the teacher model to the student model, CD encourages the student model to enhance the compactness of intra-class label-wise embeddings, which can improve its classification performance.

For a batch of examples, let $\{e_{ik}^{\mathcal{T}}\}_{i=1}^{b}$ and $\{e_{ik}^{\mathcal{S}}\}_{i=1}^{b}$ respectively denote the intra-class label-wise embeddings with respect to class $k \in [q]$ generated by the teacher and student models. Then, we can capture the structural relation between any two intra-class label-wise embeddings $\boldsymbol{e}_{ik}^{\mathcal{T}}$ and $\boldsymbol{e}_{jk}^{\mathcal{T}}$

by measuring their distance in the embedding space:

$$\phi_{\text{CD}}(\boldsymbol{e}_{ik}^{\mathcal{T}}, \boldsymbol{e}_{jk}^{\mathcal{T}}) = \begin{cases} \|\boldsymbol{e}_{ik}^{\mathcal{T}} - \boldsymbol{e}_{jk}^{\mathcal{T}}\|_2 & y_{ik} = 1, y_{jk} = 1, \\ 0 & \text{otherwise.} \end{cases} \quad (4)$$

It is worth to note that we only consider the structural relation between any two valid label-wise embeddings, *i.e.*, the embeddings with respect to positive labels. Similar to Eq.(4), for any two intra-class label-wise embeddings $\boldsymbol{e}_{ik}^{\mathcal{S}}$ and $\boldsymbol{e}_{jk}^{\mathcal{S}}$, we can obtain the structural relation $\phi_{\text{CD}}(\boldsymbol{e}_{ik}^{\mathcal{S}}, \boldsymbol{e}_{jk}^{\mathcal{S}})$ for the student model.

By enforcing the teacher and student structural relations to maintain the consistency for each pair of intra-class label-wise embeddings, we can achieve the class-aware structural consistency as follows:

$$\mathcal{L}_{\text{LED-CD}} = \sum_{k=1}^{q} \sum_{i,j \in [b]} \ell(\phi_{\text{CD}}(\boldsymbol{e}_{ik}^{\mathcal{T}}, \boldsymbol{e}_{jk}^{\mathcal{T}}), \phi_{\text{CD}}(\boldsymbol{e}_{ik}^{\mathcal{S}}, \boldsymbol{e}_{jk}^{\mathcal{S}})), \quad (5)$$

where $\ell$ is a function to measure the consistency between the teacher and student structural relations. In experiments, we use the following Huber loss function as a measurement:

$$\ell(a, b) = \begin{cases} \frac{1}{2}(a-b)^2 & |a-b| \le 1, \\ |a-b| - \frac{1}{2} & \text{otherwise.} \end{cases} \quad (6)$$

where $a$ and $b$ are two different structural relations.

### 3.2.2 INSTANCE-AWARE LABEL-WISE EMBEDDING DISTILLATION

Instance-aware label-wise embedding distillation (ID) aims to improve the distillation performance by exploring the structural relations among inter-class label-wise embeddings from the same image. Generally, one can hardly distinguish between two different semantic objects occurring in an image due to the high similarities they share. For example, see the right side of Figure 3, for an image annotated with *sky*, *sea* and *beach*, one can hardly distinguish between the semantic objects *sky* and *sea*, since they share the same color and similar texture. A feasible solution is to exploit other useful information, such as their spatial relation, *i.e.*, *sky* is always above *sea*. Due to the powerful learning capacity, the large model is able to distinguish between the similar semantic objects by exploiting such implicit supervised information. From the perspective of learned feature representations, as shown in the right side of 3, the teacher model tends to learn a dispersed structure of inter-class label-wise embedding, which is beneficial for improving its discrimination ability. By distilling the structural knowledge from the teacher model, ID enforces the student model to enhance the dispersion of inter-class label-wise embeddings, which can improve its discrimination ability.

For a given instance $\boldsymbol{x}_i$, let $\{\boldsymbol{e}_{ik}^{\mathcal{T}}\}_{k=1}^{q}$ and $\{\boldsymbol{e}_{ik}^{\mathcal{S}}\}_{k=1}^{q}$ respectively denote the label-wise embeddings generated by teacher and student models. Then, we can capture the structural relation between any two inter-class label-wise embeddings $\boldsymbol{e}_{ik}^{\mathcal{T}}$ and $\boldsymbol{e}_{il}^{\mathcal{T}}$ by measuring their distance in the embedding space:

$$\phi_{\text{ID}}(\boldsymbol{e}_{ik}^{\mathcal{T}}, \boldsymbol{e}_{il}^{\mathcal{T}}) = \begin{cases} \|\boldsymbol{e}_{ik}^{\mathcal{T}} - \boldsymbol{e}_{il}^{\mathcal{T}}\|_2 & y_{ik} = 1, y_{il} = 1, \\ 0 & \text{otherwise.} \end{cases} \quad (7)$$

Note that in Eq.(7), we only consider the structural relation between any two valid label-wise embeddings, *i.e.*, the embedding with respect to positive labels. Similar to Eq.(7), for any two inter-class label-wise embeddings $\boldsymbol{e}_{ik}^{\mathcal{S}}$ and $\boldsymbol{e}_{il}^{\mathcal{S}}$, we can obtain the structural relation $\phi_{\text{ID}}(\boldsymbol{e}_{ik}^{\mathcal{S}}, \boldsymbol{e}_{il}^{\mathcal{S}})$ for the student model.

By encouraging the teacher and student model to maintain the consistent structure of intra-instance label-wise embeddings, we can minimize the following $\mathcal{L}_{\text{LED-ID}}$ loss to achieve the instance-aware structural consistency where $\ell(.)$ is Huber loss as defined in Eq.(6):

$$\mathcal{L}_{\text{LED-ID}} = \sum_{i=1}^{b} \sum_{k,l \in [q]} \ell(\phi_{\text{ID}}(\boldsymbol{e}_{ik}^{\mathcal{T}}, \boldsymbol{e}_{il}^{\mathcal{T}}), \phi_{\text{ID}}(\boldsymbol{e}_{ik}^{\mathcal{S}}, \boldsymbol{e}_{il}^{\mathcal{S}})). \quad (8)$$

Finally, the overall objective function of L2D (Eq.(2)) can be re-written as follows:

$$\mathcal{L}_{\text{L2D}} = \mathcal{L}_{\text{BCE}} + \lambda_{\text{MLD}}\mathcal{L}_{\text{MLD}} + \lambda_{\text{LED-CD}}\mathcal{L}_{\text{LED-CD}} + \lambda_{\text{LED-ID}}\mathcal{L}_{\text{LED-ID}}, \quad (9)$$

where $\lambda_{\text{MLD}}$, $\lambda_{\text{LED-CD}}$ and $\lambda_{\text{LED-ID}}$ are all balancing parameters.

Table 1: Results on MS-COCO where teacher and student models are in the **same** architectures.

| Teacher | RepVGG-A2 | | | ResNet-101 | | | WRN-101 | | | Swin-S | | |
|---|---|---|---|---|---|---|---|---|---|---|---|---|
| Student | RepVGG-A0 | | | ResNet-34 | | | WRN-50 | | | Swin-T | | |
| Metrics | mAP | OF1 | CF1 | mAP | OF1 | CF1 | mAP | OF1 | CF1 | mAP | OF1 | CF1 |
| Teacher | 72.71 | 74.11 | 68.63 | 73.62 | 73.89 | 68.61 | 74.70 | 75.56 | 70.73 | 81.70 | 80.48 | 77.12 |
| Student | 70.02 | 72.49 | 66.77 | 70.31 | 72.49 | 66.82 | 74.45 | 75.43 | 70.61 | 79.59 | 79.18 | 75.42 |
| RKD | 70.08 | 72.39 | 66.73 | 70.13 | 72.44 | 66.78 | 74.70 | 75.71 | 70.84 | 79.63 | 79.19 | 75.57 |
| PKT | 70.11 | 72.47 | 66.80 | 70.43 | 72.64 | 66.68 | 74.54 | 75.47 | 70.58 | 79.64 | 79.09 | 75.39 |
| ReviewKD | 70.00 | 72.35 | 66.82 | 70.39 | 72.62 | 66.76 | 74.03 | 75.29 | 70.36 | 79.81 | 79.18 | 75.55 |
| MSE | 70.26 | 72.54 | 66.99 | 70.54 | 72.75 | 66.85 | 74.53 | 75.60 | 70.71 | 79.67 | 79.20 | 75.52 |
| PS | 70.65 | 72.89 | 67.60 | 70.86 | 72.66 | 67.12 | 75.12 | 76.05 | 71.63 | 79.96 | 79.64 | 76.20 |
| MLD | 70.74 | 72.81 | 67.46 | 70.68 | 72.69 | 67.19 | 74.92 | 75.75 | 71.21 | 80.11 | 79.68 | 76.44 |
| L2D | **72.81** | **74.59** | **69.49** | **72.87** | **74.45** | **69.43** | **76.61** | **77.08** | **72.79** | **81.59** | **81.03** | **77.86** |

Table 2: Results on MS-COCO where teacher and student models are in the **different** architectures.

| Teacher | ResNet-101 | | | Swin-T | | | ResNet-101 | | | Swin-T | | |
|---|---|---|---|---|---|---|---|---|---|---|---|---|
| Student | RepVGG-A0 | | | ResNet-34 | | | MobileNet v2 | | | MobileNet v2 | | |
| Metrics | mAP | OF1 | CF1 | mAP | OF1 | CF1 | mAP | OF1 | CF1 | mAP | OF1 | CF1 |
| Teacher | 73.62 | 73.89 | 68.61 | 79.43 | 78.77 | 75.07 | 73.62 | 73.89 | 68.61 | 79.43 | 78.77 | 75.07 |
| Student | 70.02 | 72.49 | 66.77 | 70.31 | 72.49 | 66.82 | 71.85 | 73.59 | 68.26 | 71.85 | 73.59 | 68.26 |
| RKD | 70.08 | 72.35 | 66.72 | 70.00 | 72.34 | 66.64 | 71.76 | 73.68 | 68.40 | 71.74 | 73.68 | 68.37 |
| PKT | 69.99 | 72.35 | 66.56 | 70.26 | 72.39 | 66.82 | 71.88 | 73.60 | 68.35 | 71.84 | 73.76 | 68.37 |
| ReviewKD | 70.00 | 72.33 | 66.62 | 70.29 | 72.39 | 66.58 | 71.92 | 73.73 | 68.48 | 71.73 | 73.71 | 68.36 |
| MSE | 70.07 | 72.50 | 66.85 | 70.33 | 72.57 | 66.72 | 71.91 | 73.68 | 68.28 | 71.80 | 73.74 | 68.38 |
| PS | 70.30 | 72.61 | 67.10 | 70.94 | 72.93 | 67.57 | 72.11 | 73.89 | 68.42 | 72.42 | 74.14 | 68.94 |
| MLD | 70.48 | 72.77 | 67.10 | 71.14 | 72.99 | 67.63 | 72.17 | 73.84 | 68.52 | 72.35 | 74.10 | 68.91 |
| L2D | **72.14** | **74.08** | **68.78** | **73.42** | **74.97** | **70.20** | **73.24** | **74.85** | **69.72** | **74.21** | **75.72** | **70.87** |

## 4 EXPERIMENTS

**Datasets.** We perform experiments on three benchmark datasets Pascal VOC2007 (Everingham et al., 2015) (VOC for short), MS-COCO2014 (Lin et al., 2014) (MS-COCO for short) and NUS-WIDE (Chua et al., 2009). VOC contains 5,011 images in the train-val set, and 4,952 images in the test set. It covers 20 common objects, with an average of 1.6 labels per image. MS-COCO contains 82,081 training images and 40,137 test images. It covers 80 common objects, with an average of 2.9 labels per image. NUS-WIDE contains 161,789 training images and 107,859 test images. It covers 81 visual concepts, with an average of 1.9 labels per image.

**Metrics.** Following existing works (Zhang & Zhou, 2013; Liu et al., 2021a; Sun et al., 2022), we adopt the mean average precision (mAP) over all classes, overall F1-score (OF1) and average per-class F1-score (CF1) to evaluate the performance. We choose OF1 and CF1, since they consider both recall and precision and thus are more comprehensive.

**Methods.** To validate the proposed method, we compare it with the following KD methods: RKD (Park et al., 2019), which captures the relations among instances to guide the training of the student model; PKT (Passalis & Tefas, 2018), which measures KL divergence between features by treating them as probability distributions; ReviewKD (Chen et al., 2021), which transfers knowledge across different stages instead of just focusing on features in the same levels; MSE (Xu et al., 2022a), which minimizes the MSE loss between logits of teacher and student model; PS (Song et al., 2021), which minimizes KL divergence of logits after a partial softmax function.

**Implementation Details.** We use the models pretrained on ImageNet (Deng et al., 2009) as the backbones. We resize the resolution of all images to $224 \times 224$ and set the batch size as 64. For each training image, we adopt a weak augmentation consisting of random horizontal flipping and a strong augmentation consisting of Cutout (Devries & Taylor, 2017) and RandAugment Cubuk et al. (2020). We use the Adam optimization method (Kingma & Ba, 2015) to train the model for 80 epochs. The one-cycle policy is used with a maximal learning rate of $0.0001$ and the weight decay (Loshchilov & Hutter, 2018) of $0.0001$. For all experiments, we set $\lambda_{\text{MLD}} = 10$, $\lambda_{\text{LED-CD}} = 100$, and $\lambda_{\text{LED-ID}} = 1000$. Parameter sensitivity analysis in Appendix D shows that the performance of

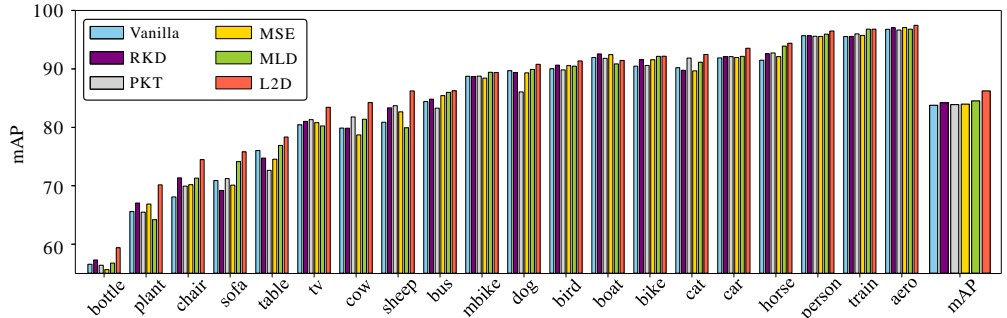

Figure 4: Comparison results of the comparing methods on VOC in terms of AP and mAP (%), where the backbones of teacher and student model are respectively ResNet-50 and ResNet-18.

L2D are insensitive to all of our balancing parameters. For the comparing methods, we set their parameters as suggested in the original papers. Especially, for all feature-based methods, we just deploy them on the feature maps which is output from the visual backbone $f$. All the experiments are conducted on GeForce RTX 2080 GPUs. More details about the used models and implementation of the label-wise embedding encoder are attached in Appendix A.

## 4.1 COMPARISON RESULTS

Table 1 and Table 2 report comparison results on MS-COCO with the same and different architectures of student and teacher models. From Table 1, it can be observed that: 1) Conventional feature-based distillation methods only achieve minor improvements in performance when compared with the student model (without distillation). This indicates these methods do not work in multi-label scenarios due to their disability to capture multiple semantics occurred in feature maps. 2) MLD can outperform conventional feature-based distillation methods in most cases, which indicates by performing one-versus-all reduction, the logits-based distillation can be adapted into multi-label knowledge distillation. 3) The proposed L2D significantly outperforms all other methods and achieves comparable performance with the teacher model, which convincingly validates the effectiveness of the proposed label-wise embedding distillation. From Table 2, we can see: 1) Compared with the same architecture, the performance gap between teacher and student model is larger for different architectures, which indicates the corresponding distillation task is harder. 2) Our method significantly outperforms all the comparing methods by a significant margin in all cases. 3) L2D achieves the best performance in all cases and significantly outperforms MLD. These results provide a strong empirical validation for the effectiveness of the proposed method.

Figure 4 illustrates the performance of the proposed methods and other comparing methods on VOC in terms of AP and mAP. It is noteworthy that the performance is ranked in descending according to the performance of student model. From the figure, it can be observed that: 1) Our proposed L2D achieves the best performance and significantly outperforms the comparing methods in terms of mAP. 2) L2D consistently achieves superior performance to the comparing methods in most classes and the performance gap is large especially for classes that the student model achieves poor performance. This observation discloses that L2D improves the performance of hard classes by enhancing the distinctiveness of feature representations. These experimental results demonstrate the practical usefulness of the proposed method. More results on VOC and NUS-WIDE can be found in Appendix B and Appendix C.

## 4.2 ABLATION STUDY

In this section, to further analyze how the proposed method improves distillation performance, Table 3 reports results of ablation studies on MS-COCO (teacher: ResNet-101, student: ResNet-34) and VOC (teacher: ResNet-50, student: ResNet-18). The first line in Table 3 reports the performance of student model without knowledge distillation. It can be observed that the distillation performance is improved slightly by only conducting multi-label logits distillation. Compared with MLD, label-wise embeddings distillation achieves the major improvement for the proposed method. It can

Table 3: Ablation studies on MS-COCO and VOC.

| MLD | CD | ID | MS-COCO | | | Pascal VOC 2007 | | |
|-----|-----|-----|-----|-----|-----|-----|-----|-----|
| | | | mAP | OF1 | CF1 | mAP | OF1 | CF1 |
| | | | 70.31 | 72.49 | 66.82 | 84.01 | 83.60 | 79.42 |
| ✓ | | | 70.68 | 72.69 | 67.19 | 84.48 | 84.07 | 80.29 |
| ✓ | ✓ | | 71.91 | 73.74 | 68.76 | 85.65 | 85.64 | 81.94 |
| ✓ | | ✓ | 71.79 | 73.62 | 68.43 | 84.59 | 84.19 | 80.24 |
| ✓ | ✓ | ✓ | 72.87 | 74.45 | 69.43 | 85.71 | 85.70 | 82.11 |

| (a) Vanilla | (b) PKT | (c) MLD | (d) L2D |

Figure 5: The differences between correlation matrices of student and teacher predicted probabilities on MS-COCO. The student distilled by our method shows significant matching between student's and teacher's correlations compared to others.

be observed that by performing CD and ID, the mAP performance achieves 1.6% and 1.48% increments, respectively. Finally, we also examine the combination of these techniques. By incorporating these components together, the fusing method achieves the best performance and significantly outperforms each other method. These results demonstrate that all of three components are of great importance to the performance of the proposed L2D.

### 4.3 DISTILLING INTER-CLASS CORRELATIONS

As discussed in the previous works (Hinton et al., 2015), the conventional supervised learning losses, *e.g.*, BCE loss, often neglects the correlations among class predictions. However, the label correlation is a foundational element for multi-label classification. Distillation losses utilize "soft targets", which can effectively distill such correlation information from the teacher model, leading to a desirable distillation performance. To validate whether the correlations can be captured by L2D effectively, Figure 5 illustrates the differences between correlation matrices of student and teacher predicted probabilities on MS-COCO. From the figure, it can be observed that: 1) Without knowledge distillation, the teacher and student correlations are very different, which indicates teacher model tends to capture a more precise correlation than student model and thus achieves better performance. 2) The representative comparing method PKT shows a large difference, which discloses that the conventional KD methods are ineffective to capture the correlations in multi-label scenarios. 3) MLD shows a reduced difference, which indicates that the rich informativeness of logits is beneficial for capturing precise correlations. 4) L2D shows significant matching between the teacher and student correlations. By enhancing the distinctiveness of label-wise embeddings, L2D can obtain more correct predicted probabilities, leading to a more precise correlation estimation.

## 5 CONCLUSION

The paper studies the problem of multi-label knowledge distillation. In the proposed method, the multi-label logits distillation explores the informative semantic knowledge compressed in the teacher logits to obtain more semantic supervision. Furthermore, the label-wise embedding distillation exploits the structural knowledge from label-wise embeddings to learn more distinctive feature representations. Experimental results on benchmark datasets validate the effectiveness of the proposed method. In future, we plan to improve the performance of MLKD by exploiting other abundant structural information.

## 6 REPRODUCIBILITY

We list the parameters for running the proposed algorithm in Section 4. The used neural networks and implementation details of the label-wise embedding encoder can be found in Appendix A. The source code and a detailed description about the code are attached in the supplementary material.

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

## A  MORE DETAILS OF IMPLEMENTATION

In order to validate the proposed method with diverse architectures, we employ some commonly used models, including ResNet (He et al., 2016), Wide ResNet (WRN) (Zagoruyko & Komodakis, 2016b), RepVGG (Ding et al., 2021), Swin Transformer (Liu et al., 2021b), and MobileNet v2 (Sandler et al., 2018). For all the backbones, we utilize their pre-trained version on the ImageNet (Deng et al., 2009) as our base model.

For all experiments, similar to the previous work (Ridnik et al., 2021b), we employ the label-wise embedding encoder consisting of a cross-attention module and a feed-forward fully-connected layer (Vaswani et al., 2017). The cross-attention module takes full queries and feature maps as the input. We assign a query per class to ensure that each query corresponds to a single semantic. The multi-label classifier $h(.)$ is a fully-connected layer for each class, which outputs a predicted logit for a class label based on the input label-wise embedding.

In experiments shown in Figure 1, for distillation cross the same architectures, we use Swin-S (Liu et al., 2021b) as the teacher and Swin-T as the student; for distillation cross different architectures, we use ResNet-101 (He et al., 2016) as the teacher and MobileNet v2 (Sandler et al., 2018) as the student.

## B  MORE RESULTS ON PASCAL VOC 2007

Table 4 and Table 5 report comparison results on Pascal VOC 2007 with the same and different architectures of student and teacher models. From the tables, it can be observed that the proposed L2D significantly outperforms all comparing methods, which convincingly validates the effectiveness of the proposed label-wise embeddings distillation. Compared with the results on MS-COCO, the performance gap between L2D and comparing methods seems to become smaller. One possible reason is that VOC only contains about 1.5 labels per image, which leads conventional KD methods to obtain a better performance.

Table 4: Results on Pascal VOC 2007 validation teacher and student models are in the **same** architectures.

| Teacher | RepVGG-A2 | | | ResNet-50 | | | WRN-101 | | | Swin-S | | |
|---|---|---|---|---|---|---|---|---|---|---|---|---|
| Student | RepVGG-A0 | | | ResNet-18 | | | WRN-50 | | | Swin-T | | |
| Teacher | 86.20 | 85.63 | 82.62 | 86.73 | 84.92 | 81.21 | 88.00 | 87.03 | 83.72 | 92.75 | 91.05 | 88.82 |
| Student | 83.79 | 83.36 | 79.83 | 84.01 | 83.60 | 79.42 | 88.52 | 87.21 | 84.08 | 91.31 | 89.98 | 88.00 |
| RKD | 83.75 | 83.41 | 79.85 | 84.48 | 83.54 | 79.83 | 88.21 | 87.33 | 84.55 | 91.52 | 90.44 | 88.51 |
| PKT | 83.63 | 83.53 | 80.04 | 84.12 | 83.10 | 79.31 | 87.69 | 87.07 | 84.14 | 91.28 | 90.17 | 88.03 |
| ReviewKD | 83.87 | 83.98 | 80.54 | 83.71 | 83.01 | 79.25 | 88.23 | 87.13 | 84.20 | 91.45 | 90.17 | 88.06 |
| MSE | 84.02 | 83.67 | 79.94 | 84.23 | 83.16 | 79.29 | 88.04 | 86.49 | 83.57 | 91.06 | 89.99 | 87.66 |
| PS | 83.77 | 83.74 | 80.28 | 84.44 | 83.78 | 79.95 | 88.30 | 86.92 | 83.91 | 91.21 | 90.25 | 88.12 |
| MLD | 83.65 | 83.66 | 80.02 | 84.48 | 84.07 | 80.29 | 88.29 | 87.16 | 84.25 | 91.43 | 90.72 | 88.81 |
| L2D | **84.56** | **84.37** | **80.82** | **85.71** | **85.70** | **82.11** | **89.52** | **88.25** | **85.69** | **91.92** | **91.34** | **89.58** |

Table 5: Results on Pascal VOC 2007 validation where teacher and student models are in the **different** architectures.

| Teacher | ResNet-50 | | | Swin-T | | | ResNet-50 | | | Swin-T | | |
|---|---|---|---|---|---|---|---|---|---|---|---|---|
| Student | RepVGG-A0 | | | ResNet-18 | | | MobileNet v2 | | | MobileNet v2 | | |
| Teacher | 86.73 | 84.92 | 81.21 | 91.43 | 89.81 | 87.63 | 86.73 | 84.92 | 81.21 | 91.43 | 89.81 | 87.63 |
| Student | 83.79 | 83.36 | 79.83 | 84.01 | 83.60 | 79.42 | 86.12 | 85.01 | 81.76 | 86.12 | 85.01 | 81.76 |
| RKD | 84.26 | 84.29 | 80.70 | 83.27 | 83.05 | 79.55 | 86.22 | 84.97 | 81.76 | 85.68 | 85.31 | 81.57 |
| PKT | 83.93 | 83.79 | 80.03 | 83.45 | 83.25 | 79.64 | 86.10 | 84.84 | 81.66 | 85.67 | 85.22 | 81.68 |
| ReviewKD | 84.07 | 83.62 | 80.34 | 83.37 | 83.08 | 78.93 | 85.87 | 85.04 | 81.73 | 85.69 | 85.10 | 81.56 |
| MSE | 84.01 | 84.05 | 80.52 | 83.60 | 83.06 | 79.46 | 86.20 | 84.94 | 81.84 | 85.80 | 85.51 | 81.98 |
| PS | 84.80 | 84.46 | 81.13 | 83.97 | 83.75 | 79.86 | 86.26 | 85.47 | 82.06 | 86.07 | 85.73 | 82.39 |
| MLD | 85.07 | 84.91 | 81.55 | 84.61 | 84.26 | 80.78 | 86.38 | 85.67 | 82.43 | 86.11 | 85.98 | 82.55 |
| L2D | **86.26** | **85.85** | **82.55** | **85.87** | **85.67** | **82.17** | **87.32** | **86.48** | **83.26** | **87.37** | **86.88** | **83.68** |

## C MORE RESULTS ON NUS-WIDE

Table 6 reports comparison results on NUS-WIDE with the same and different architectures of student and teacher models. For distillation between the same architectures, we choose a ResNet-101 (He et al., 2016) as the teacher and a ResNet-34 as the student. For distillation between different architectures, we choose a Swin-T (Liu et al., 2021b) as the teacher and a MobileNet v2 (Sandler et al., 2018) as the student. From the tables, it can be observed that the proposed L2D significantly outperforms all comparing methods, which convincingly validates the effectiveness of the proposed label-wise embeddings distillation.

Table 6: Results on NUS-WIDE validation.

| Teacher | ResNet-101 | | | Swin-T | | |
|---|---|---|---|---|---|---|
| Student | ResNet-34 | | | MobileNet v2 | | |
| Metrics | mAP | OF1 | CF1 | mAP | OF1 | CF1 |
| Teacher | 55.32 | 75.56 | 61.31 | 59.73 | 77.30 | 65.44 |
| Student | 53.41 | 75.10 | 60.08 | 54.49 | 75.72 | 61.74 |
| RKD | 53.62 | 75.20 | 59.91 | 54.76 | 75.69 | 61.74 |
| PKT | 53.55 | 75.08 | 60.35 | 54.59 | 75.69 | 61.74 |
| ReviewKD | 53.52 | 75.23 | 60.44 | 54.85 | 75.84 | 61.75 |
| MSE | 53.52 | 75.13 | 59.94 | 54.86 | 75.80 | 61.69 |
| PS | 54.14 | 75.43 | 60.79 | 55.18 | 75.91 | 62.35 |
| MLD | 54.44 | 75.36 | 60.73 | 55.36 | 76.00 | 62.52 |
| L2D | **55.31** | **76.17** | **62.79** | **56.91** | **76.92** | **63.89** |

## D PARAMETER SENSITIVITY ANALYSIS

In this section, we study the influence of balancing parameters $\lambda_{\text{MLD}}$, $\lambda_{\text{LED-CD}}$ and $\lambda_{\text{LED-ID}}$ on the performance of L2D. A commonly used setting of the hyperparameter in vanilla KD that balances KL divergence against CE is 0.9 (Tian et al., 2019), which means the balancing parameter for CE is 0.1 and the one for KL divergence is 0.9. So we choose 10 for the balancing parameter for MLD, which is closest to the setting of vanilla KD. We set the balancing parameter for LED-ID larger than LED-CD considering that the LED-ID may carry less information because there are only less than 3 labels for a instance on average, though it seems unnecessary since parameter sensitivity experiments in Figure 6 show that the performance of L2D are not sensitive to all of our balancing parameters.

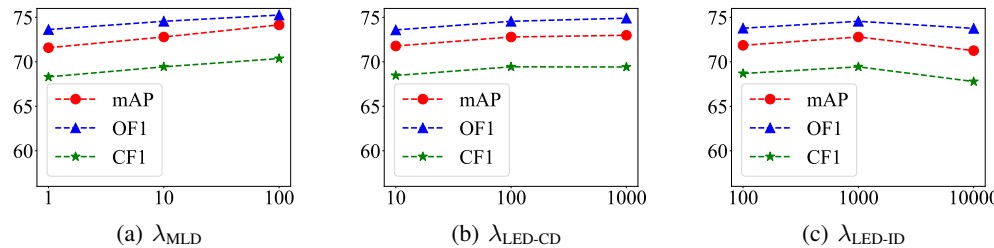

Figure 6: Student models' performance comparisons with different values of $\lambda_{\text{MLD}}$, $\lambda_{\text{LED-CD}}$ and $\lambda_{\text{LED-ID}}$ respectively on MS-COCO with a ResNet-101 as the teacher and a ResNet-34 as the student.

## E VISUALIZATION OF ATTENTION MAPS

To further show the effectiveness of our proposed method L2D, we visualize some attention maps of the penultimate layer in the visual backbones using LayerCAM (Jiang et al., 2021) implemented

by Fernandez (2020). We compare attention maps of the student model trained by L2D with some other methods in Figure 7 8 9. We compare L2D with: 1)Vanilla: student trained without distillation; 2)PKT: a classical feature-based method. In each figure, the first column shows the raw picture and the other columns show class activation maps overlaying on the raw picture. Each row represents a certain class. From these figures, we can find that L2D can locate the specified object more precisely than the other methods, which means it can not only pay attention to target objects, but also resist interference from similar but unrelated objects. All these comparisons show that L2D outperforms all comparing methods. It validates the effectiveness of our proposed label-wise embeddings distillation and shows great potential in MLKD.

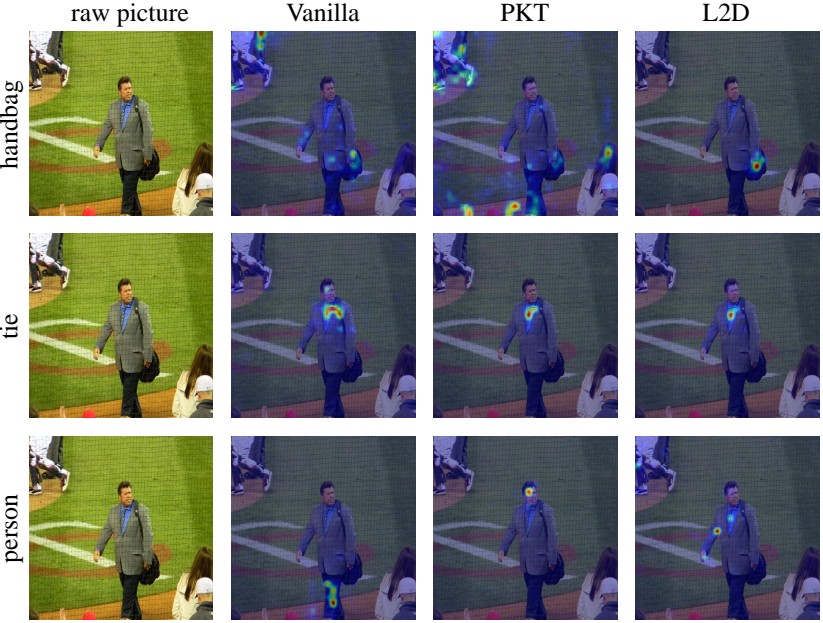

Figure 7: An example of visualization of attention maps. We can find that on attention head for class *handbag*, both Vanilla and PKT are interfered by some other objects and do not pay all attention to the handbag, but our L2D resists such interference successfully.

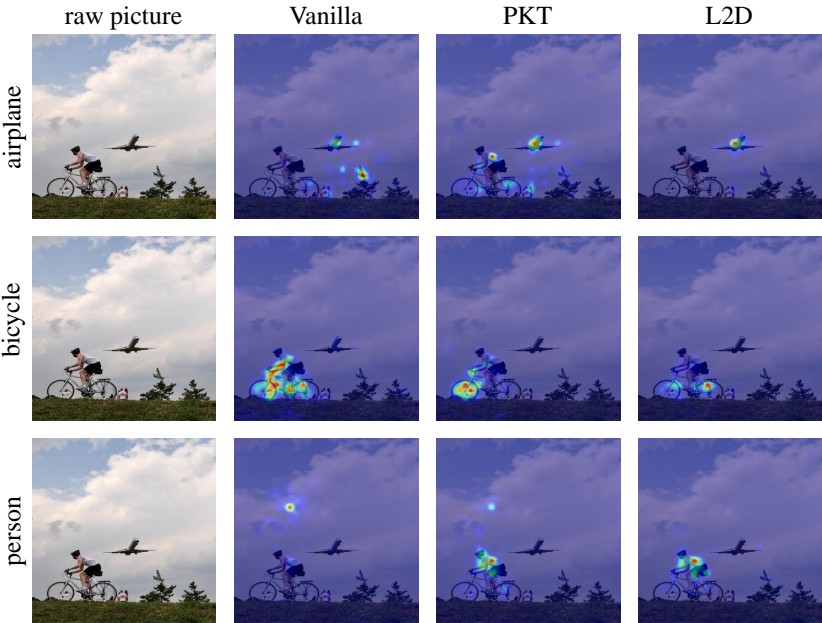

Figure 8: An example of visualization of attention maps. We can find that on attention head for class *airplane*, both Vanilla and PKT do not pay all attention to the airplane: Vanilla is interfered by the plant and PKT is interfered by the boy. But our L2D resists such interference successfully. On attention head for class *person*, both Vanilla and PKT are interfered by the shades on the cloud, but L2D is not.

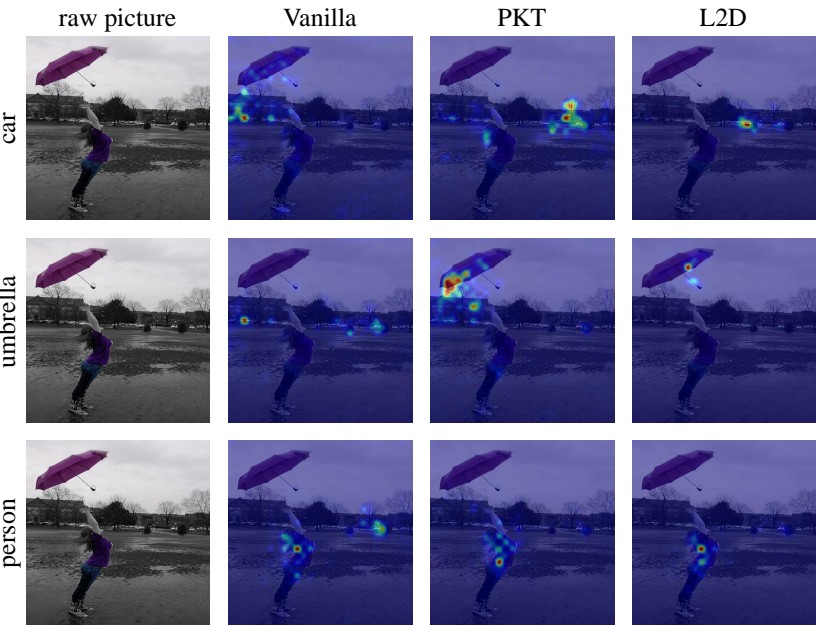

Figure 9: An example of visualization of attention maps. We can observe that on attention heads for class *car* and class *person*, Vanilla pays some of its attention to unrelated objects. On attention head for class *umbrella*, PKT pays some of its attention to the house. Only L2D can concentrate on these three targets precisely.

