# OpenReview forum: "Multi-Label Knowledge Distillation"
_ICLR.cc/2023/Conference — Submitted to ICLR 2023_

### Official Review · Reviewer_SHPH · 2022-10-24

**Confidence:** 4
**Correctness:** 3
**Technical Novelty And Significance:** 2
**Empirical Novelty And Significance:** 2
**Recommendation:** 3

**Clarity, Quality, Novelty And Reproducibility:**

The paper is writing clearly and easy to follow. The class/instance-aware embedding distillation loss seems common in other areas so not sure if this counts as a novel contribution.

**Strength And Weaknesses:**

Strength:
1. The approach is simple and easy to implement.
2. The experiments result shows the improvement over the previous method on resolution 224x224.
3. The code is released for reproducing the results.

Weakness:
1. Although the title is Multi-Label Knowledge Distillation (MLKD), and as stated in first contribution this paper proposes a general MLKD framework, it is only about multi label image classification.
2. The experiment part only covers VOC and COCO with 224x224 resolutions. However, previous work [1] [2] mainly working on 448x448, and thus it is hard to compare and validate the result of this work. Also it would be better to include the result on NUS-WIDE.
3. There seems not enough novelty by only adding a intra-class/instance embedding distillation loss.

**Summary Of The Paper:**

This paper proposes a distillation method on multi label image classification task. It introduces a multi label logits distillation task and intra-class/instance embedding distillation loss. And performs experiments on COCO and VOC dataset demonstrating the effectiveness of the
proposed method. The source code is provided for reproducibility.

**Summary Of The Review:**

This paper is well-written and the experiments result shows the improvement over the previous method on certain setting. But more experiments are needed to show the improvements over previous methods. Also there is a concern about the novelty.

---

> ### Author Response · Authors · 2022-11-18
> **Response to Reviewer SHPH**
>
> Dear Reviewer,
>
> Thank you very much for the constructive comments.
>
> > Although the title is Multi-Label Knowledge Distillation (MLKD), and as stated in first contribution this paper proposes a general MLKD framework, it is only about multi label image classification.
>
> Thanks for your suggestion. Following previous KD works [1,2,3], we focus on image classification, since the task is the most typical application of multi-label learning. In future, we will apply the proposed method to other realistic tasks, such NLP.
>
> > The experiment part only covers VOC and COCO with 224x224 resolutions. However, previous work [1] [2] mainly working on 448x448, and thus it is hard to compare and validate the result of this work. Also it would be better to include the result on NUS-WIDE.
>
> Thanks for your advice! Since our goal is to validate the effectiveness of the proposed method, we adopt the most commonly used setting, 224x224 resolution. Similar results can be obtained with 448x448 resolutions.
>
> The results on NUS are illustrated as follows. More results on this dataset could be found in the revised version.
>
> | Teacher & Student | Res101 & Res34 | Res101 & Res34 | Res101 & Res34 | Swin-T & MBNv2 | Swin-T & MBNv2 | Swin-T & MBNv2 |
> | ----------------- | -------------- | -------------- | -------------- | -------------- | -------------- | -------------- |
> | Metrics           | mAP            | OF1            | CF1            | mAP            | OF1            | CF1            |
> | Teacher           | 55.32          | 75.56          | 61.31          | 59.73          | 77.30          | 65.44          |
> | Student           | 53.41          | 75.10          | 60.08          | 54.49          | 75.72          | 61.74          |
> | RKD               | 53.62          | 75.20          | 59.91          | 54.76          | 75.69          | 61.74          |
> | PKT               | 53.55          | 75.08          | 60.35          | 54.59          | 75.69          | 61.74          |
> | ReviewKD          | 53.52          | 75.23          | 60.44          | 54.85          | 75.84          | 61.75          |
> | MSE               | 53.52          | 75.13          | 59.94          | 54.86          | 75.80          | 61.69          |
> | PS                | 54.14          | 75.43          | 60.79          | 55.18          | 75.91          | 62.35          |
> | MLD               | 54.44          | 75.36          | 60.73          | 55.36          | 76.00          | 62.52          |
> | L2D               | **55.31**      | **76.17**      | **62.79**      | **56.91**      | **76.92**      | **63.89**      |
>
> > There seems not enough novelty by only adding a intra-class/instance embedding distillation loss.
>
>  Our main contributions are summarized as follows:
>
> 1. There exists a large gap between MLKD and KD, since in multi-label learning, each instance is associated with multiple semantics. This can be further validated by our experiments, where traditional KD methods for single-label classification do not work for MLKD problems. We propose MLKD framework to formalize the problem, with the goal of improving the distillation performance of the student model in multi-label-scenarios.
> 2. The proposed label-wise embedding distillation is significant from the previous KD methods that often perform distillation on image-level feature representations. In multi-label scenarios, for any two images that share the same object while still have their individual objects, it is often hard to measure their similarity based on the image-level feature representations. In contrast, we perform distillation on label-wise embeddings , which are able to precisely measure the similarities between any two objects.
> 3. The designed two embedding distillation losses leverage intra-class and intra-instance structural information of the teacher model to learn more compact feature representations for the student model.
>
>
>
> All of revisions are highlighted in the revised version.
>
> Please don’t hesitate to let us know for any additional comments. Thank you!
>
>
>
> [1] Wonpyo Park, Dongju Kim, Yan Lu, and Minsu Cho. Relational knowledge distillation. In *Proceedings of the IEEE/CVF Conference on Computer Vision and Pattern Recognition*, pp. 3967–3976, 2019.
>
> [2] Yonglong Tian, Dilip Krishnan, and Phillip Isola. Contrastive representation distillation. In *International Conference on Learning Representations*, 2019.
>
> [3] Pengguang Chen, Shu Liu, Hengshuang Zhao, and Jiaya Jia. Distilling knowledge via knowledge review. In *Proceedings of the IEEE/CVF Conference on Computer Vision and Pattern Recognition*, pp. 5008–5017, 2021.

---

### Official Review · Reviewer_c6wM · 2022-10-24

**Confidence:** 5
**Clarity, Quality, Novelty And Reproducibility:** Please refer to my detailed comments …
**Correctness:** 3
**Technical Novelty And Significance:** 2
**Empirical Novelty And Significance:** 2
**Recommendation:** 3

**Strength And Weaknesses:**

Strength:

(1) The organization is good, which makes the paper easy to follow.

(2) The method is simple, and the results on several datasets seem good.

Weaknesses:

(1) The presentation could be improved. For example, the abbreviation "MLL" lacks explanation. The meaning of the red line in Figure 1 is also unclear, and I guess it is the result of the teacher.

(2) For the BCE loss, it is commonly used for multi-label classification. The simple distillation of MLD for this predicted probability is not novel. Moreover, the structural relation by class-aware label-wise embedding distillation and instance-aware label-wise embedding distillation is similar to the global graph and local graph-based similarity consistency. In this case, this paper can be regarded as a combination of existing multi-label learning and KD methods, and the overall novelty is not satisfying. Though the authors present several different distance measurements, the overall contribution is still insufficient.

(3) The authors mainly compare with traditional KD methods. Please explain how these methods deal with the multi-class situation in the paper.

(4) I would like to see a detailed comparison with existing multi-label methods and KD methods in the loss function, respectively.

(5) According to the results in Tables 1 and 2, the L2D even achieves much better results than the teacher. Please explain this phenomenon.

**Summary Of The Paper:**

This paper presents the multi-label knowledge distillation to address the issue of multiple semantic labels in multi-label learning scenarios. MKD and LED are introduced for network learning. Experiments on several datasets demonstrate its superiority.

**Summary Of The Review:**

My main concern lies in the novelty, since both multi-label classification and KD have been well studied, and this paper does not provide some impressive new design for this task.

---

> ### Author Response · Authors · 2022-11-18
> **Response to Reviewer c6wM**
>
> Dear Reviewer,
>
> Thank you very much for the constructive comments.
>
> > The presentation could be improved. For example, the abbreviation "MLL" lacks explanation. The meaning of the red line in Figure 1 is also unclear, and I guess it is the result of the teacher.
>
> Thanks for your suggestion. We have added the explanation of the abbreviation "MLL" in the revised version. The red dashed lines represent the performance of teacher model, which have already been explained in the fifth line of the caption.
>
> > For the BCE loss, it is commonly used for multi-label classification. The simple distillation of MLD for this predicted probability is not novel. Moreover, the structural relation by class-aware label-wise embedding distillation and instance-aware label-wise embedding distillation is similar to the global graph and local graph-based similarity consistency. In this case, this paper can be regarded as a combination of existing multi-label learning and KD methods, and the overall novelty is not satisfying. Though the authors present several different distance measurements, the overall contribution is still insufficient.
>
> There are significant differences between our work and previous methods. Our main contributions are summarized as follows:
>
> 1. There exists a large gap between MLKD and KD, since in multi-label learning, each instance is associated with multiple semantics. This can be further validated by our experiments, where traditional KD methods for single-label classification do not work for MLKD problems. We propose MLKD framework to formalize the problem, with the goal of improving the distillation performance of the student model in multi-label-scenarios.
> 2. The proposed method performs embedding distillation on label-level feature representations, which is significantly different form previous works that utilize global graph and local graph-based similarity consistency on image-label feature representations. In multi-label scenarios, for any two images that share the same object while still have their individual objects, it is often hard to measure their similarity based on the image-level feature representations. In contrast, performing label-wise embedding distillation is more reasonable, since each label-wise embedding corresponds a single label.
>
> > The authors mainly compare with traditional KD methods. Please explain how these methods deal with the multi-class situation in the paper.
>
>  Thank you for pointing this out. We have added implementation details of traditional KD methods for multi-label scenarios in Section 4 in the revised version..
>
> > I would like to see a detailed comparison with existing multi-label methods and KD methods in the loss function, respectively.
>
>  It is noteworthy that our goal is to improve the distillation performance of the student model instead of the performance of multi-label learning. To validate effectiveness of the proposed method, we compare it with the state of the art KD methods in Table 1,2,4,5,6 and conduct extensive ablation studies in Table 3.
>
> > According to the results in Tables 1 and 2, the L2D even achieves much better results than the teacher. Please explain this phenomenon.
>
> This phenomenon would occur when the gap of model capabilities between the student and the teacher is small. Similar situations have been happened in tradition KD methods. For example, in Table 1 and 2 in [1], it can be found that ReviewKD can help the student perform outperform the teacher in the following settings on CIFAR-100: WRN40-2 --> WRN16-2, VGG13 --> VGG8, WRN40-2 --> ShuffleNetV1.
>
>
>
> All of revisions and the existing statement about red dashes in Figure 1 are highlighted in the revised version.
>
> Please don’t hesitate to let us know for any additional comments. Thank you!
>
>
>
> [1] Chen, Pengguang, et al. "Distilling knowledge via knowledge review." *Proceedings of the IEEE/CVF Conference on Computer Vision and Pattern Recognition*. 2021.

---

### Official Review · Reviewer_fqJx · 2022-10-25

**Confidence:** 4
**Correctness:** 4
**Technical Novelty And Significance:** 4
**Empirical Novelty And Significance:** 4
**Recommendation:** 5

**Clarity, Quality, Novelty And Reproducibility:**

The proposed method is clear and novel. The quality of this paper looks favorable overall.

**Strength And Weaknesses:**

Strength
- The authors tackle a relatively new problem, knowledge distillation for multi-label classification, and they propose a method suitable to address the problem of the existing approaches.
- The proposed method is straightforward and shows favorable performance.


Weakness
- This work is not the first work to apply the knowledge distillation technique to the multi-label classification model. As the authors mentioned, Song et al. and Xu et al. also apply knowledge distillation to multi-label classification models. There are several more works with similar frameworks such as [1]. In the introduction, the authors should not only introduce the “distillation for multi-label classification” but also explain what the problem of the existing frameworks is and how the proposed method addresses the problem. The authors should explain why the proposed method shows favorable performance.
[1] Multi-label image classification via knowledge distillation from weakly-supervised detection. ACM MM 2018.
- The proposed method itself is simple and straightforward. Therefore, it would be better to further analyze the design choices of the proposed model to claim the impact of the proposed method. For example, why is the loss function in Eqn (6) used? Is there any better option for the authors to try for the loss function?
- It would be better to have more experimental results in the main paper. For example, instead of the results for the VOC dataset in Fig. 4, it would be more informative to show the result with different architectures like Tables 1 and 2. Also, the results on the NUS-WIDE dataset, which is a famous multi-label classification dataset, are missing.
- Finally, it would be better to have an analysis of the CAM before and after applying the proposed method, which is popularly used in knowledge distillation literature. For the analysis of Fig. 5, it is hard to see the correlation due to the large class number. I suggest analyzing the VOC dataset to better visualize the correlation between classes.






**Summary Of The Paper:**

The authors propose a multi-label knowledge distillation method. On the one hand, it exploits the logit's informative semantic knowledge by label decoupling with the one-versus-all reduction strategy. On the other hand, it enhances the distinctiveness of the learned feature representations by leveraging the structural information of label-wise embeddings.

**Summary Of The Review:**

Although it would be better to add several more results in the main paper, this paper addresses an interesting problem, and the performance of the proposed model seems favorable.


-----after rebuttal-----

After reading the comments from the other reviewers, I think the novelty of the proposed method is not strong enough.
Therefore, I decided to reduce the score to 'below threshold.'

---

> ### Author Response · Authors · 2022-11-18
> **Response to Reviewer fqJx (1/2)**
>
> Dear Reviewer,
>
> Thank you very much for the constructive comments.
>
> > This work is not the first work to apply the knowledge distillation technique to the multi-label classification model. As the authors mentioned, Song et al. and Xu et al. also apply knowledge distillation to multi-label classification models. There are several more works with similar frameworks such as [1]. In the introduction, the authors should not only introduce the “distillation for multi-label classification” but also explain what the problem of the existing frameworks is and how the proposed method addresses the problem. The authors should explain why the proposed method shows favorable performance.
> >
> > [1] Multi-label image classification via knowledge distillation from weakly-supervised detection. ACM MM 2018.
>
> Thanks for pointing out the missing reference. We have already added it in the revised revision.
>
> There are significant differences between our framework and previous works [1,2,3]. Previous works employ vanilla KD techniques to improve performance of multi-label learning, while our work is the first time to study how to improve the KD performance of the student model in multi-label scenarios. We have already done some discussions in the last paragraph of Section 2 and we will add some more explanations in the revised version.
>
> > The proposed method itself is simple and straightforward. Therefore, it would be better to further analyze the design choices of the proposed model to claim the impact of the proposed method. For example, why is the loss function in Eqn (6) used? Is there any better option for the authors to try for the loss function?
>
> We use Huber loss in experiments, since it is commonly used to measure difference between two relations [4]. We will add some experiments to analyze other loss functions in the future version.
>
> > It would be better to have more experimental results in the main paper. For example, instead of the results for the VOC dataset in Fig. 4, it would be more informative to show the result with different architectures like Tables 1 and 2. Also, the results on the NUS-WIDE dataset, which is a famous multi-label classification dataset, are missing.
>
> Thanks for your suggestion. Results on NUS-WIDE are shown as below and have been added in the revised version.
>
> | Teacher & Student | Res101 & Res34 | Res101 & Res34 | Res101 & Res34 | Swin-T & MBNv2 | Swin-T & MBNv2 | Swin-T & MBNv2 |
> | ----------------- | -------------- | -------------- | -------------- | -------------- | -------------- | -------------- |
> | Metrics           | mAP            | OF1            | CF1            | mAP            | OF1            | CF1            |
> | Teacher           | 55.32          | 75.56          | 61.31          | 59.73          | 77.30          | 65.44          |
> | Student           | 53.41          | 75.10          | 60.08          | 54.49          | 75.72          | 61.74          |
> | RKD               | 53.62          | 75.20          | 59.91          | 54.76          | 75.69          | 61.74          |
> | PKT               | 53.55          | 75.08          | 60.35          | 54.59          | 75.69          | 61.74          |
> | ReviewKD          | 53.52          | 75.23          | 60.44          | 54.85          | 75.84          | 61.75          |
> | MSE               | 53.52          | 75.13          | 59.94          | 54.86          | 75.80          | 61.69          |
> | PS                | 54.14          | 75.43          | 60.79          | 55.18          | 75.91          | 62.35          |
> | MLD               | 54.44          | 75.36          | 60.73          | 55.36          | 76.00          | 62.52          |
> | L2D               | **55.31**      | **76.17**      | **62.79**      | **56.91**      | **76.92**      | **63.89**      |
>
>
> Due to the page limitation, we move the table of average precision (AP) on VOC into the appendix.

---

> ### Author Response · Authors · 2022-11-18
> **Response to Reviewer fqJx (2/2)**
>
> > Finally, it would be better to have an analysis of the CAM before and after applying the proposed method, which is popularly used in knowledge distillation literature. For the analysis of Fig. 5, it is hard to see the correlation due to the large class number. I suggest analyzing the VOC dataset to better visualize the correlation between classes.
>
> Thank you very much for your suggestion! An analysis of the CAM has been added in the revised version.
>
> It is noteworthy that capturing the correlation among labels on COCO is far more important than that on VOC, since the average number of labels is very small on VOC. From Fig.5, it can be observed that the difference is significant reduced by using our method, which can be reflected by a much paler color than comparing methods.
>
>
>
> All of revisions are highlighted in the revised version.
>
> Please don’t hesitate to let us know for any additional comments. Thank you!
>
>
>
> [1] Yongcheng Liu, Lu Sheng, Jing Shao, Junjie Yan, Shiming Xiang, and Chunhong Pan. Multi-label
> image classification via knowledge distillation from weakly-supervised detection. In *Proceedings
> of the 26th ACM International Conference on Multimedia*, pp. 700–708, 2018.
>
> [2] Jiazhi Xu, Sheng Huang, Fengtao Zhou, Luwen Huangfu, Daniel Zeng, and Bo Liu. Boosting
> multi-label image classification with complementary parallel self-distillation. In *Proceedings of
> the Thirty-First International Joint Conference on Artificial Intelligence*, 2022a.
>
> [3] Liangchen Song, Jialian Wu, Ming Yang, Qian Zhang, Yuan Li, and Junsong Yuan. Handling
> difficult labels for multi-label image classification via uncertainty distillation. In *Proceedings of
> the 29th ACM International Conference on Multimedia*, pp. 2410–2419, 2021.
>
> [4] Wonpyo Park, Dongju Kim, Yan Lu, and Minsu Cho. Relational knowledge distillation. In *Proceedings of the IEEE/CVF Conference on Computer Vision and Pattern Recognition*, pp. 3967–3976, 2019.

---

### Official Review · Reviewer_gMao · 2022-10-26

**Confidence:** 3
**Correctness:** 4
**Technical Novelty And Significance:** 3
**Empirical Novelty And Significance:** 3
**Recommendation:** 5

**Clarity, Quality, Novelty And Reproducibility:**

This paper clearly introduces the motivation and the proposed models. The training parameters are introduced in the paper. The novelty is fine.

**Strength And Weaknesses:**

Strength:

The background and the motivation of the setting is well-introduced. The motivation of the work is reasonable and logical.

The proposed modules including the label decoupling and the embedding-based lobal correlation learning are clearly introduced with motivation and explanations. The motivation is rational and the explanation is reasonable.

Experiments demonstrate the effectiveness of the proposed model.

Weaknesses:

Embedding based label structural knowledge extraction/learning is a general approach for multi-label learning. To this end, there are one module which could be considered as a novel module. More explanations about why/how the embedding strategy is novel could be introduced.


**Summary Of The Paper:**

This paper proposed a novel method for knowledge distillation in multi-label learning scenarios. Multi-label learning aims to solve the problem where multiple positive labels exists in a given single sample. However, general knowledge distillation methods mainly focus on single-label learning scenario while ignoring the latent label correlations/structural knowledge across the labels or the label embedding.

This work proposed a specific design framework which preserves the label structural knowledge in knowledge distillation training process. More specifically, a label decoupling with the one-versus all reduction strategy is deployed, and label-wise embedding is further explored to get the label structural knowledge across different label. Experimental results demonstrate the effectiveness of the proposed model.


**Summary Of The Review:**

This paper proposed a novel approach for multi-label knowledge distillation. The proposed modules are reasonable with high performance.

---

> ### Author Response · Authors · 2022-11-18
> **Response to Reviewer gMao**
>
> Dear Reviewer,
>
> Thank you very much for the constructive comments.
>
> > Embedding based label structural knowledge extraction/learning is a general approach for multi-label learning. To this end, there are one module which could be considered as a novel module. More explanations about why/how the embedding strategy is novel could be introduced.
>
> Thanks for your advice! The proposed label-wise embedding distillation is significantly different from the previous KD methods that often perform distillation on image-level feature representations. In multi-label scenarios, for any two images that share the same object while still have their individual objects, it is often hard to measure their similarity based on the image-level feature representations. In contrast, we perform distillation on label-wise embeddings, which are able to precisely measure the similarities between any two objects. The designed two embedding distillation losses leverage intra-class and intra-instance structural information of the teacher model to learn more compact feature representations for the student model.
>
>
>
> All of revisions are highlighted in the revised version.
>
> Please don’t hesitate to let us know for any additional comments. Thank you!

---

> > ### Comment · Reviewer_gMao · 2022-11-27
> > **Followup**
> >
> > I have carefully read the reviews from other reviewers and the response from the authors.
> >
> > First, thanks for authors' response the extra explanations of the model.
> >
> > Second, I also agree with the opinions of other reviewers that the novelty of the proposed model is still not good enough. As discussed above, the label-embedding and label correlation learning (even in multi-label scenario) is a general strategy. In addition, as mentioned by reviewer c6wM, graph related strategies share the same approach as the multi-label relation learning module.
> >
> > To this end, I host negative opinion of this submission.

---

### Official Review · Reviewer_NCc1 · 2022-11-04

**Confidence:** 3
**Correctness:** 3
**Technical Novelty And Significance:** 2
**Empirical Novelty And Significance:** 2
**Recommendation:** 3

**Clarity, Quality, Novelty And Reproducibility:**

The paper is well-written and easy to understand. The major novelty comes from the two proposed label-wise embedding distillation approaches, i.e., LED-CD and LED-ID, which leverage the multi-label info to implement the relation-based knowledge distillation. However, the experiments in the paper are insufficient to justify the effectiveness of the proposed approach.

**Strength And Weaknesses:**

# Strength
* The paper is well-written and easy to understand.

# Weakness
* The literature survey for knowledge distillation in this paper is incomplete. The survey only covers two KD categories (logit-based and feature-based) and misses the relation-based category [1]. This miss is critical as the major novelty of this paper comes from proposing losses in this third KD category.
* The proposed KD approach requires labeled data, which limits its application. I would question if Eq. (4) and (7) need to consider only the distances between positive label embeddings. It may also work if we simply include the embedding distances for all label pairs in the loss – disregarding whether the labels are positive or not. Would be good to have experiments verifying the design decision for Eq. (4) and (7).
* Some claims in the paper lack supports from the experiments. For example, the authors claim the LED-ID loss would leverage spatial relations between labels. It would be good to have experimental results supporting such a claim.
* Some experimental setup lacks explanation, e.g., how are the weights for MLD, LED-CD, and LED-ID losses determined as 10, 100, and 1000 in this paper?
* The baselines are considerably weak as they are all with a single type of knowledge distillation, while the proposed L2D combines multiple distillation losses (i.e., BCE, MLD, LED-CD, and LED-ID). The proposed L2D loss may have the best results simply because of the loss combination (a known approach for performance improvement). A fair baseline should also be a system combining multiple losses.



[1] Gou, J., Yu, B., Maybank, S.J. et al. Knowledge Distillation: A Survey. Int J Comput Vis 129, 1789–1819 (2021). https://doi.org/10.1007/s11263-021-01453-z  (available at https://arxiv.org/pdf/2006.05525.pdf )

**Summary Of The Paper:**

This paper proposes doing knowledge distillation (KD) for multi-label ML models via KD approaches at both logit and feature representation levels.  The major novelty comes from the two proposed label-wise embedding distillation approaches, i.e., LED-CD and LED-ID, which leverage the multi-label info to implement the relation-based knowledge distillation. The experimental results, however, are insufficient for justifying or providing insights into the proposed approach.

**Summary Of The Review:**

Though the paper proposes an interesting idea of label-wise embedding distillation leveraging the multi-label info, the experimental setup in the paper, however, makes it difficult to justify the effectiveness of the proposed distillation method. Because of that, I recommend rejecting the paper. I also suggest that the authors improve the literature survey, implement a stronger baseline, and conduct experiments verifying the loss design in Eq. (4) and (7).

---

> ### Author Response · Authors · 2022-11-18
> **Response to Reviewer NCc1 (1/2)**
>
> Dear Reviewer,
>
> Thank you very much for the constructive comments.
>
> > The literature survey for knowledge distillation in this paper is incomplete. The survey only covers two KD categories (logit-based and feature-based) and misses the relation-based category. This miss is critical as the major novelty of this paper comes from proposing losses in this third KD category.
>
> Thank you very much for the suggestion. Different from the mentioned survey, our paper simply group feature-based and relation-based methods into one category. In our paper, we have introduced as many relation-based methods as possible, including RKD, although we divide them into the feature-based category. It is noteworthy that there exist surveys [1] that divide KD into only two categories: KD from logits and KD from intermediate layers. We will make the categorization of KD method more precise in the revised version.
>
> > The proposed KD approach requires labeled data, which limits its application. I would question if Eq. (4) and (7) need to consider only the distances between positive label embeddings. It may also work if we simply include the embedding distances for all label pairs in the loss – disregarding whether the labels are positive or not. Would be good to have experiments verifying the design decision for Eq. (4) and (7).
>
> Thanks for your reminder. We only perform distillation on positive label embeddings due to the fact that only positive label embeddings carry specific semantic information in multi-label learning. For instance, a positive label embedding of class *Dog* means that there is a dog in the image, while a negative label embedding of class *Dog* cannot be associated with any certain object. Moreover, we cannot claim that there is a *Cat* or anything else with respect to a negative label embedding of class *Dog*. As a result, it is more reasonable to ignore negative label embeddings.
>
> > Some claims in the paper lack supports from the experiments. For example, the authors claim the LED-ID loss would leverage spatial relations between labels. It would be good to have experimental results supporting such a claim.
>
> Thank you for your advice. To make it easier to understand our main idea, we take spatial relation between labels as an example. In experiments, we further conduct ablation studies to validate the effectiveness of LED-ID loss as shown in Table 3.
>
> > Some experimental setup lacks explanation, e.g., how are the weights for MLD, LED-CD, and LED-ID losses determined as 10, 100, and 1000 in this paper?
>
> Similar to vanilla KD methods [2,3,4] that set the balancing parameter as 0.9 for KD loss (0.1 for CE loss), we set the balancing parameter for MLD loss as 10. Considering the number of positive labels within an instance (about 2.5 labels per instance on COCO), we set a larger balancing parameter for LED-ID than LED-CD. Furthermore, we conduct parameter sensitivity analysis in the revised version. The results in Figure 6 show that the performance is insensitive to all of the parameters.

---

> ### Author Response · Authors · 2022-11-18
> **Response to Reviewer NCc1 (2/2)**
>
> > The baselines are considerably weak as they are all with a single type of knowledge distillation, while the proposed L2D combines multiple distillation losses (i.e., BCE, MLD, LED-CD, and LED-ID). The proposed L2D loss may have the best results simply because of the loss combination (a known approach for performance improvement). A fair baseline should also be a system combining multiple losses.
>
> It is noteworthy that all comparing methods in experiments contain BCE loss. We do not adding MLD loss into comparing methods for following two reasons:
>
> 1. MLD is firstly proposed in this paper.
>
> 2. The performance of those feature-based methods after adding MLD methods is still near the performance of MLD. The results are shown as below.
>
> | Teacher      | ResNet-101 | ResNet-101 | ResNet-101 |
> | ------------ | ---------- | ---------- | ---------- |
> | Student      | ResNet-34  | ResNet-34  | ResNet-34  |
> | Metrics      | mAP        | OF1        | CF1        |
> | Teacher      | 73.62      | 73.89      | 68.61      |
> | Student      | 70.26      | 72.63      | 66.66      |
> | RKD          | 70.13      | 72.44      | 66.78      |
> | PKT          | 70.38      | 72.46      | 66.74      |
> | ReviewKD     | 70.39      | 72.62      | 66.76      |
> | MLD          | 70.64      | 72.63      | 67.27      |
> | MLD+RKD      | 70.28      | 72.49      | 66.67      |
> | MLD+PKT      | 70.27      | 72.59      | 66.74      |
> | MLD+ReviewKD | 70.59      | 72.58      | 66.75      |
>
> Also, the results of ablation study in Section 4.3 strongly proves that all the components in our proposed loss function are of vital importance.
>
>
>
> All of revisions are highlighted in the revised version.
>
> Please don’t hesitate to let us know for any additional comments. Thank you!
>
>
>
>
> [1] Wang, Lin, and Kuk-Jin Yoon. "Knowledge distillation and student-teacher learning for visual intelligence: A review and new outlooks." *IEEE Transactions on Pattern Analysis and Machine Intelligence* (2021).
>
> [2] Hinton, Geoffrey, Oriol Vinyals, and Jeff Dean. "Distilling the knowledge in a neural network." *arXiv preprint arXiv:1503.02531* 2.7 (2015).
>
> [3] Tian, Yonglong, Dilip Krishnan, and Phillip Isola. "Contrastive representation distillation." *arXiv preprint arXiv:1910.10699* (2019).
>
> [4] Chen, Pengguang, et al. "Distilling knowledge via knowledge review." *Proceedings of the IEEE/CVF Conference on Computer Vision and Pattern Recognition*. 2021.

---

> > ### Comment · Reviewer_NCc1 · 2022-12-04
> > **Follow up**
> >
> > Thanks to the authors for the response and for addressing some of the comments in the revised version. I went through the authors’ responses and the revised paper closely. Here are my further comments:
> >
> > ## Major
> > * The experiment protocol of the paper is still concerning
> >     * I appreciate that the authors make it clearer in the latest revision how the hyperparameters are selected and how the baseline/competitive systems are implemented. However, the details increase my concern regarding the experimental protocol of the paper:
> >       * I disagree with the author's statement in the new appendix “D PARAMETER SENSITIVITY ANALYSIS” saying that the performance is not sensitive to the hyperparameters. It looks like changing the hyperparameters can lead up to a 1% abs difference in performance metrics. My concern is that this indicates the authors did not really do hyperparameter tuning for the baseline/competitive systems using a validation dataset to ensure strong baseline/competitive systems. This means the experimental results in the paper might be misleading and oversell the proposed approach.
> >       * The sentence in the latest revision “Especially, for all feature-based methods, we just deploy them on the feature maps which are output from the visual backbone f” raises my concern regarding the weak baseline/competitive systems used in the paper. I would consider a more fair (and intuitive) setup to apply the distillation approaches on top of g.
> >       * I also notice the MS-COCO performance for ​​ResNet-101 in Table 1 is surprisingly low (73.62%) when compared to the numbers (>78.3%) in Table 1 in the other paper https://arxiv.org/pdf/2205.10986.pdf. This again raises my concerns regarding the experimental setup in the paper.
> >
> > ## Minor
> > * We need to better distinguish feature-based KD and relation-based KD in the paper
> >     * The authors' response that “We only perform distillation on positive label embeddings due to the fact that only positive label embeddings carry specific semantic information in multi-label learning.” shows the difference between feature-based KD and relation-based KD. For the feature-based approach, as suggested by its name, you do not really care about whether the label is positive or not and just want the student to generate similar features as the teacher model in both positive and negative cases.
> > * The added Appendix “E VISUALIZATION OF ATTENTION MAPS” in the latest revision is helpful and addresses my original comment. I appreciate that the authors considered adding such info, which helps the readers to understand the effect of the proposed method more easily.
> >
> >
> > With the major concern regarding the experimental setup, I hold my original reject decision.

---

### Decision · Program_Chairs · 2023-01-20

**Decision:**

Reject

**Justification For Why Not Higher Score:**

The technical novelty is not sufficient. Experimental results lacking in some respect.

**Justification For Why Not Lower Score:**

N/A

**Metareview: Summary, Strengths And Weaknesses:**

The authors present a knowledge distillation (KD) solution for multi-label ML models. The proposed idea exploits the informative semantic knowledge from the logits by label decoupling with the one-versus-all reduction strategy. Moreover, it enhances the distinctiveness of the learned feature representations by leveraging the structural information of label-wise embeddings. Experimental results are reported on several datasets.

The main contributions are the class-aware label-wise embedding distillation (LED-CD) and the instance-aware label-wise embedding distillation (LED-ID), which use multi-label info to put forth relation-based knowledge distillation.

The key strength of the paper, as pointed out by all reviewers, is the quality of the  presentation style along with the good organization of the content. The technical novelty appears instead to be limited/incremental, and some aspects of the contributions seem to exist in prior art (class/instance-aware embedding distillation loss seems common in other areas). The authors’ response did not changed the reviewers’ opinions. Another main problem is the experimental validation, which has actually generated additional concerns after the authors’ rebuttal ( MS-COCO performance reported by the authors for ​​ResNet-101 appear to be not reliable, and  the parameter sensitivity analysis generated some new concerns).


**Summary Of Ac-Reviewer Meeting:**

N/A